# Adaptive Routing Optimization Algorithm in Community-Oriented Opportunistic Networks for Mobile Health

**DOI:** 10.3390/s19081876

**Published:** 2019-04-19

**Authors:** Weimin Chen, Zhigang Chen, Fang Cui

**Affiliations:** 1School of Computer Science and Engineering, Central South University, Changsha 410075, China; hnyycwm@163.com (W.C.); cf2695@163.com (F.C.); 2School of Information and Electronic Engineering, Hunan City University, Yiyang 413000, China

**Keywords:** mobile health, opportunistic networks, relationship, community, message ferry

## Abstract

The appearance of a large number of mobile intelligent devices boosts the fast rise of mobile health (mHealth) application. However, due to the sensitivity and complexity of medical data, an efficient and secure mobile communication mode is a very difficult and challenging task in mHealth. The Opportunistic Networks (OppNets) is self-organizing and can expand the communication capacity by the movement of nodes, so it has a good prospect in the application of mHealth. Unfortunately, due to the shortage of stable and reliable end-to-end links, the routing protocol in OppNets has usually lower performance and is unsafe. To address these issues, we propose an adaptive routing optimization algorithm in OppNets for mHealth. This routing scheme firstly analyzes the relationship between nodes and defines the average message forwarding delay as a new metric to selectively forward messages, and then designs a local community detection algorithm based on the metric to adapt to the characteristics of OppNets, and finally resorts to some super-nodes to ferry messages between different communication domains. The simulation results demonstrate the efficiency and effectiveness of the proposed scheme. It increases the delivery ratio by about 30%, decreases delay by about 35%, and decreases the number of forwarding by about 5%, by comparing it with several existing routing schemes. We believe that the relationship between nodes, community, and message ferrying will play an important role in routing of OppNets for mHealth.

## 1. Introduction

The mobile health (mHealth) provides an effective approach for medical and health services through the use of mobile communication technologies, such as the mobile phone. The mHealth improves not only the working efficiency of the hospital but also the health of the people, and it is becoming one of the most active research and application directions in the field of medical information recently [1]. However, the flourish of mHealth still faces many challenges, including communication mode and information security. The Opportunistic Networks (OppNets) is composed of a large number of mobile intelligent terminals and takes advantage of their movement to facilitate data transmission and exchange. The appearance of OppNets makes people’s communication with each other break through the limitations of time and space, and provides them with quick and convenient access to information [2,3]. Therefore, OppNets is more suitable for the mHealth applications and has a good prospect of application [4].

Different from the traditional networks, OppNets adopts the "storage-carry-forward" routing strategies to transmit the messages [5]. The advantage of this strategy is that it can forward messages between disconnected mobile nodes and expand the communication capacity of the network. However, there are some problems, such as lower delivery ratio, longer transferring latency, and the dependency on the cooperation of the nodes. To design more efficient routing mechanism, the community is introduced into OppNets, because nodes in the group usually have more opportunities to forward messages [6,7,8].

Unfortunately, the community-based routing scheme encounters critical challenges in practical situations, such as the measure metric of the relationship between nodes, the community detection algorithm adapted to the unique signatures of OppNets, and efficient messages transmission between communication domains. To overcome these disadvantages, we propose an adaptive routing optimization algorithm in OppNets for mHealth. This algorithm firstly utilizes a new metric of node relations to detect the communities based on local information. Then, the super nodes are adaptively deployed to the network to ferry messages between communication domains.

The main work of this paper includes the following three aspects: (1) To more accurately measure the relationship strength between nodes, we define a new metric using the average message forwarding delay with the historical encounters. (2) In order to avoid the difficulty that the global information of OppNets is too hard to obtain, we present a local community detection algorithm based on the metric. (3) With the help of ferry nodes, we propose an adaptive routing optimization algorithm in OppNets for mHealth.

## 2. Related Work

As a new health care model, mHealth has attracted the attention of many researchers. They mainly introduce the current situation of the mHealth application, discuss the main problems in the process of the application, and put forward some opinions on the development trend in the future. Free et al. [9] analyzes the effectiveness of mobile technology interventions for health care consumers. Istepanian et al. [10] discusses the trends in the applications of mobile technologies for different healthcare schemes. Doukas et al. [11] designs and implements a mobile healthcare system that is based on Cloud Computing. In [12], Inglong et al. think service provisions and individual attitudes play an important role in the promotion of the mHealth Application. Rongxing et al. [13] presents a mHealth framework based on OppNets to efficiently protect the user privacy.

On the other hand, to improve routing performance of OppNets, many routing algorithms are proposed from different perspectives, such as Spray and Wait [14] and PRoPHET [15]. With the popularization of mobile intelligent terminals, a number of new applications of OppNets have been spawned, such as pocket switched networks (PSN) and vehicular networks. In these scenarios, the community-based routing algorithms are proposed. Hui et al. [16] described the community structure of PSN and proposed an algorithm, named BUBBLE, which selects nodes with high centrality value and community members of destination as relay nodes. Jianwei et al. [17] organize nodes into communities on the basis of the contact frequencies and choose more active nodes as the relays in communication. Thomas et al. [18] use group structures for efficient routing in terms of the relationship among human society. In [7], Fang et al. used a simple method to construct a new cross-group network structure. Jia et al. [19] present a data transmission scheme based on the weight distribution between nodes and community’s reconstitution in social opportunistic networks. Xingyuan et al. [20] propose an overlapping communities detection algorithm based on vital nodes and initial seeds. Xianhuan et al. [21] propose a non-overlapping community detection algorithm based on the trust relationship between users. Yeqing et al. [22] propose an effective data transmission strategy based on node socialization. It has a high packet delivery ratio by using the communities and selecting optimal next-hop nodes, but it has some disadvantages, such as the measurement of the nodes’ similarity by distance. In addition, there are some routing algorithms based on the relationship [23], context aware [24], etc. Furthermore, the routing based on message ferry is introduced in some applications of OppNets where nodes are distributed sparsely. Wenrui et al. [25] propose a message ferrying scheme and discuss the use of the ferry routing under the conditions of the constrained number of ferry nodes. The optimized way-points [26] was proposed to determine the mobile path of the ferry node by calculating the encounter probability. Xue et al. [27] select the node with the highest power as the ferry and propose the single-ferry and multiple-ferry mechanisms. Shangxing et al. [28] formulate the problem of robots ferrying messages to communicate between statically-placed sources and sink pairs. Harounabadi et al. [29] use a stigmergic communication for the coordination of ferries and the decision functions based on the local observations of a ferry, and present an adaptive on-the-fly decision maker for multi-ferry delay tolerant networks.

As presented above, the mHealth system need a wireless ad-hoc network that can efficiently and securely transmit massive and sensitive medical information. However, the flooding-based routing scheme does not consider the heterogeneity of nodes and simply relies on multiple copies to transmit messages. The message ferry-based routing scheme can improve the transmission performance in OppNets where nodes are distributed sparsely. The routing based on communities provides an effective measure to promote the communication ability of network, but it has some defects to transmit messages efficiently between communities. Therefore, it is necessary to propose a new approach to achieve the reliable and efficient data transmission in OppNets for mHealth.

## 3. Adaptive Routing Optimization Algorithm

### 3.1. Network Model

In our study, the OppNets for mHealth is designed as shown in Figure 1. The network consists of two different types of mobile nodes including ordinary and ferry nodes. The ordinary node is a common mobile intelligent terminal with several pieces of software that can send or accept messages and record logs of conversations, and the embedded communication device such as Bluetooth. It can be divided into different communities according to a certain metric, for example, the relationship between nodes. Compared to the ordinary node, the ferry node can reach a longer communication distance and don’t need to join the community.

The transmission mechanism of the network meets the following rules: (1) The forwarding of a message in the community is dependent on the ordinary nodes, while the communication between communities is carried out by the ferry nodes. (2) An encounter node is chosen as the relay node if it satisfies a certain condition, such as being more active than the source node, one of the ferry nodes, and the same community with the destination node.

### 3.2. Analysis of Node Relations

In OppNets for mHealth, nodes are human-carried intelligent devices, and their movement and contacts are carried out by the carriers. Hence, they are heterogeneous and have social relationships and behaviors of people, which are used to analyze their mobility and predict their contacts for reliable message delivery.

#### 3.2.1. Historical Encounters

The historical encounters of nodes can be successfully collected with the wireless terminal devices after a period of network running. They usually include information such as the number, duration, and time interval of encounters, and may also include the node’s position, speed, and direction of movement. Hence, we can obtain the movement laws of nodes over a period of time and measure the strength of the relationship between nodes. For example, the more frequent the contact between pairs of nodes are, the closer their relationship is.

In the previous researches [16,30,31,32,33], there are some metrics which can be used to analyze the relationship between nodes, such as the number of encounters, the frequency of encounters, and the total or average duration of encounters. Unfortunately, they have some shortcomings and cannot exactly evaluate the relationship between nodes in different situations. Therefore, we define a new metric based on the historical encounters as follows:(1)$$w_{ij}=\frac{T}{\int_{0}^{T}f(t)dt}$$
where, *w_ij_* is the measure metric and denotes the reciprocal of the average delay of messages forwarding in the time *T*. *f* (*t*) is a function and denotes the forwarding delay. The message is forwarded at once during the encounter, while it can only wait for the next encounter in other cases. Therefore, we have:(2)$$f(t)=\{\begin{array}{cc} 0 & \mathrm{during}\mathrm{the}\mathrm{encounter}\\ t_{next}-t & \mathrm{otherwise} \end{array}$$

Furthermore, the interval time of encounters between nodes *v_i_* and *v_j_* is *t_k_*, and *k* =1, 2,⋯, *n*. Then, we specify the metric as follows:(3)$$w_{ij}=\frac{2T}{t_{1}^{2}+t_{2}^{2}+\cdots +t_{n}^{2}}=\frac{2T}{\sum_{k=1}^{n}t_{k}^{2}}$$

Clearly, the larger *w_ij_* is, the closer the relationship between node *v_i_* and *v_j_* is.

#### 3.2.2. Social Features

The social features of nodes can be represented by physical features and logical features of a person (for example, membership in an organization, such as age, gender, blood group, and members of the organization. Li et al. in [29] present that the more the same social features of two nodes is, the more frequent their contact is. Therefore, we think the social features can also reflect a certain relationship between nodes.

The 2006 Infocom datasets [34] provides 10 social features based on questionnaires, and they are nationality, studies, languages, affiliation, position, city, country, topics, and so on. The MIT reality mining datasets [35] provide, also, several social features, such as neighborhood, daily commute, hangouts, working hour, affiliation, and research group. In these social features, only a small part of them have a significant influence on the relationship between nodes, and they can be obtained by computing their Shannon entropy as follows:(4)$$E(f_{i})=-\sum_{j=1}^{k_{j}}p(x_{j})\log(x_{j})$$
where, *f_i_* is the *i*th social features of nodes, *x_j_* is a value of *f_i_*, and *p*(*x_j_*) is the probability of *x_j_*. Obviously, if the value of *E*(*f _i_*) is bigger, the social feature *f_i_* is more important.

To measure the relationship between nodes based on the social features, we used the number of their social features in common as follows:(5)$$F_{ij}=n_{ij}$$

Obviously, if the value of the metric *F_ij_* is bigger, node *v_i_* and *v_j_* is closer.

#### 3.2.3. Proposed New Metric

Both the social feature and the historical encounter can reflect the relationship between nodes, but they are different from a logical point of view. The former is stable and can be obtained according to the investigation before network deployment, and it is related to the social relationship and physical features of the carrier, while the latter is dynamic and calibrated and maintained at scheduled time intervals, and it depends on the physical position of nodes. Therefore, they are complementary and can be combined to use as a hybrid relationship as follows:(6)$$m_{ij}=\alpha \times \frac{w_{ij}-w_{\min}}{w_{\max}-w_{\min}}+(1-\alpha )\times \frac{F_{ij}-F_{\min}}{F_{\max}-F_{\min}}$$
where, *w_ij_* is the new metric which is combined historical encounters with social features of nodes by normalized processing, α is an adjustment parameter to control the influence of historical encounters and social features on the relationship between nodes.

### 3.3. Community Detection

Community detection is a complex and difficult task. Researchers proposed many community detection algorithms [36], such as Laplace matrix spectral bisection algorithm [37], Kernighan-Lin algorithm [38], GN algorithm [39], and the fast algorithm [40].

In the actual application of OppNets for mHealth, the global attribute of the network is usually very hard to get due to the network partition. On the other side, some nodes may belong to multiple different communities. Therefore, the community detection algorithm of OppNets should meet two conditions: it is based on the local attribute of the network and can deal with the overlapping communities. For this reason, we present a local community detection algorithm on the basis of the relationship between nodes. Its details are as follows:

Step 1. Compute the active value of nodes: Considering a large degree of nodes is often the center of a community, we simplify the computation of the active value as follows:(7)$$C_{i}=\frac{\sum_{j\neq i}m_{ij}}{N_{i}}$$
where, *C**_i_* is the active value of node *v_i_*, *m_ij_* is the proposed metric, and *N_i_* is the number of nodes which have the close relationship with *v_i_*.

Step 2. Select the core nodes: Sorting by the active value, the nodes with the larger value are picked out. Considering some of the selected node are intimate and may belong to the same community, they cannot be a core node and should be removed. Hence, we have:(8)$$S_{ij}=\frac{\left|\Gamma_{i}\cap \Gamma_{j}\right|}{\left|\Gamma_{i}\cup \Gamma_{j}\right|}$$
where, Γ*_i_* is the intimate node set of *v_i_*, |Γ*_i_* ∩ Γ*_j_*| is the number of the common intimate nodes between *v_i_* and *v_j_*, *S**_ij_* is the similarity value between node *v_i_* and *v_j_*. If *S**_ij_* is greater than a threshold (e.g., 0.8), *v_i_* and *v_j_* is thought to be similar, and one of them is removed.

Step 3. Expand community according to the fitness degree: The fitness degree of node *v_i_* for community *C* can be defined as: (9)$$F_{i}=\frac{K_{in}(C+i)}{{\left(K_{in}(C+i)+K_{out}(C+i)\right)}^{\alpha }}-\frac{K_{in}(C)}{{\left(K_{in}(C)+K_{out}(C)\right)}^{\alpha }}$$
where *K_in_*(*C*) is the metric sum of internal node in community *C*, *K_out_*(*C*) is the metric sum between *C* and other community, *α* is a regulatory factor which can control the size of the community, and *C* + *i* is node *v_i_* join *C*. If *F**_i_* > 0, *v_i_* is allowed to join *C*.

Step 4. Combine redundant communities: (10)$$O_{1}{}_{2}=\frac{\left|C_{1}\cap C_{2}\right|}{min(\left|C_{1}\right|,\left|C_{2}\right|)}$$
where, |*C*_1_| is the number of nodes in *C*_1_, |*C*_1_ ∩ *C*_2_| is the number of identical nodes in *C*_1_ and *C*_2_, *O*_12_ is the overlap value between *C*_1_ and *C*_2_. If *O*_12_ is bigger than a value (for example, 0.75), *C*_1_ and *C*_2_ are redundant and need to be combined into a new community.

Step 5. Repeat Steps 3 to 4 above for the rest of nodes until each node is added in a community.

### 3.4. Deployment of Ferry Nodes

The community can be introduced into the OppNets for mHealth to improve their routing performance. However, there is a communication bottleneck between communities because nodes are very rare and inactive in these areas. Considering the role of ferry nodes, this section will discuss how to deploy the ferry nodes in the community boundary.

The network topology of OppNets is dynamically changing because of node movement. In order to achieve adaptive deployment of ferry nodes, we introduce the virtual potential field theory to control the movement of ferry nodes. In the virtual potential field theory [41], there is a pair of forces between two objects in the vacuum as follows: (11)$$\vec{F_{12}}=-\vec{F_{21}}=K\times \frac{M_{1}M_{2}}{D^{2}}\times \vec{D_{0}}$$
where, *K* is a coefficient, *M* is the mass of the object, *D* is the distance of two objects, $\vec{D_{0}}$ is the direction of the force.

Just like the objects in the vacuum, the ferry nodes in OppNets will be subject to the repulsion and attraction forces of the neighbor nodes. Furthermore, we improve the Formula (11) as follows: First, we simplify the calculation of forces and set *K* = 1, *M*_1_ = 1, *M*_2_ = 1, namely, $\vec{F}=\frac{\vec{D_{0}}}{D^{2}}$. Next, we only consider the effect of force upon the ferry nodes. Then, the repulsion or attraction forces of the ferry nodes are determined by their position, not their distance. Specifically, we consider the two cases as follows:

(1) Ferry node is located in a community.

In this case, the ferry node should be away from the center of the community and move to the edge. Therefore, it will be excluded by the nodes located in the inner layer of the community, and be attracted to the outer layer nodes. To achieve this goal, the following two problems need to be addressed: First, the layer-number of a node located in the community can be determined. It is obtained by tagging, in the process of community detection, based on the breadth-first search algorithm. For example, the layer-number of the center node in the community is determined to be 0, and its neighbor nodes are 1, and so on. Second, the position of a node relative to the ferry node in the community can be determined. We take the reference layer to be *n_r_* = (*n*_1_ + *n*_2_)/2, where *n*_1_ and *n*_2_ are the two layers’ numbers of nodes that have nearer distance from the ferry node. If the layer number of a node *n* > *n_r_*, it is located in the inner layer of the community, and it is attracted to the ferry node. Otherwise, it is in the outer layer and excludes the ferry node. The forces diagram of ferry nodes located in a community is shown in Figure 2A. In the perceptual boundary of the ferry node *fn*, there are four ordinary nodes *n*_21_, *n*_22_, *n*_31_, *n*_32_. Where *n*_21_ and *n*_22_ are ordinary nodes in the second layer of the community, and *n*_31_ and *n*_32_ are ordinary nodes in the third layer of the community. There are two repulsion forces, *f*_21_ and *f*_22_, and two attraction forces, *f*_31_ and *f*_32_. The resultant force is as follows:(12)$$\vec{f}=\vec{f_{21}}+\vec{f_{22}}+\vec{f_{31}}+\vec{f_{32}}$$
Under the action of this resultant force, the ferry node moves to the edge of the community.

(2) Ferry node is located in the area between communities.

In this case, the ferry node should be excluded by all its neighbor nodes, including ordinary nodes in communities and other ferry nodes, and it should move to the center area between the communities. The force diagram of ferry nodes located in the area between communities is shown in Figure 2B. In the perceptual boundary of the ferry node *fn*_1_, there are two ordinary nodes, *n_a_* and *n_b_*, and two ferry nodes, *fn*_2_ and *fn*_3_. Where *n_a_* is in the *a*th community and *n_b_* is in the *b*th community. There are four repulsion forces, *f_a_*, *f_b_*, *f_2_,* and *f_3_*. The resultant force is as follows:(13)$$\vec{f^{\prime }}=\vec{f_{a}}+\vec{f_{b}}+\vec{f_{2}}+\vec{f_{3}}$$
Under the action of this resultant force, the ferry node moves to the center area between the communities.

### 3.5. Transmission Algorithm 

In OppNets, the transmission scheme of messages is based on the greed strategy in general. That is, compared to the current node with a message, an intermediate node has more opportunities to access the destination node. Then it is selected as a relay and can receive the message forwarded by the current node. In this paper, the network contains a number of communities. The messages are forwarded by ordinary nodes within the community, and are ferried by ferry nodes between communities.

The transmission strategy in the community is: If the intermediate node *v_i_* is met, then the following requirement applies: *v_i_* is one of the ferry nodes or *w_i_* > *w_s_* (where, *w_i_* is the metric sum of relationships between *v_i_* and other nodes in the community). In the case whereby *v_i_* is thought to be more active than the source node *v_s_* and it has more opportunities to access the destination node, *v_s_* forwards the message to *v_i_* and *v_i_* becomes a relay node. The transmission strategy between the communities is: If the intermediate node *v_i_* is in the ordinary node and is a member in the community of the destination node, the ferry node *v_f_* forwards the message to *v_i_* and *v_i_* becomes a relay node.

The Algorithm 1 explains the process of the adaptive routing. At a particular time, the node *v_s_* needs to send a message *m* to the node *v_d_* and encounters a node *v_i_*. If *v_i_* is *v_d_* or *v_i_* has more opportunities to access *v_d_* than *v_s_*, *v_i_* is selected as a relay and receives *m* from *v_s_*. Otherwise, *v_s_* continues carrying *m* until it meets the next node. Then, *v_i_* with *m* also participates in the forwarding, and this process does not stop until *v_d_* accepts *m*.


**Algorithm 1 An Adaptive Routing Optimization Algorithm**
**Input:** source node *v_s_*, destination node *v_d_*, ferry node *v_f_*, message *m*, community C, metric *w*1: **while**
*v_d_* without *m*
**do**2:  current node *v_c_* (*v_s_*, *v_f_* or relay node *v_r_*) encounters node *v_i_* without *m*;3:  **if**
*v_i_* is *v_d_*
**then**4:   *v_i_* accepts *m* from *v_c_* and the forwarding process ends;5:  **else**6:   **if**
*v_c_* and *v_d_* are in the same C **then**7:    **if**
*v_i_* is in the C and *w_id_* > *w_cd_*
**then**8:     *v_i_* is selected as *v_r_*, and *v_c_* forwards *m* to *v_i_*;9:    **end if**10:   **else**11:    **if**
*v_c_* is *v_f_*
**then**12:     **if**
*v_i_* and *v_d_* are in the same C **then**13:      *v_i_* is selected as *v_r_*, and *v_c_* forwards *m* to *v_i_*;14:     **end if**15:    **else**16:     **if**
*v_i_* is *v_f_* or *w_id_* > w_cd_
**then**17:      *v_i_* is selected as *v_r_*, and *v_c_* forwards *m* to *v_i_*;18:     **end if**19:     **end if**20:   **end if**21:  **end if**22: **end while**

## 4. Simulations

To evaluate the performance of the proposed routing in OppNets for mHealth, a few simulation experiments were done with several different types of contact traces.

### 4.1. Simulation Setting

#### 4.1.1. Simulation Data

In all our simulations, the following data set was used. (1) Realistic contact trace: We used the datasets which were collected with iMotes in Infocom 2006 [34]. It is publicly available from the CRAWDAD collection. There were 79 students and researchers who had 74,981 contacts in 337,418 s. In our simulation, the data of 61 participants was only used, because other people’s information was incomplete. Six social features were extracted from these datasets, including graduated school, topics, nationality, country, city, and affiliation. The first 40 h of data was used to gather the historical data for the network training, and the remaining 53 h of data was used to evaluate the proposed routing performance. (2) Synthetic contact trace: It was formed in the simulator ONE. The parameter of ONE was: The size of the network map was 12 × 12 km; the wireless range of the ferry nodes were around 300 m and their movement speed was 20 m/s; the wireless range of the ordinary nodes was around 150 m and their movement speed was 10 m/s; the simulation lasted 26,500 s.

#### 4.1.2. Performance Metrics

In all simulations, the following metrics of routing performance were used: (1) Delivery Ratio: It is the average percentage of the number of packets that successfully achieved the destination and that were sent from by the source node over a certain period of time; (2) Delivery Delay: It is the average transmission time of the packets that are accepted by the destination; (3) Hop Count: It is the average number of the relay when packets are delivered successfully; (4) Number of Forwarding: It is the average number of packets during the forwarding process.

#### 4.1.3. Methods

In our simulations, the following routing schemes were used: (1) Enhanced community-based routing with ferry node (ECRF): Our proposed routing scheme. (2) Enhanced community-based routing (ECR): ECRF without ferry nodes. (3) Community-based routing (CR) [32]: In this scheme, an encounter node is forwarded the message if it is within the community of the destination node. (4) Space filling way-point (SFWP) [25]: The scheme divided the whole network i into grids, and deployed ferry nodes in the center of grids. In addition, it used the ECR algorithm to forward messages. (5) Spray-and-wait (S-W) [14]: In the first phase, the current node gives half of the message replicas to the encountered node; in the second phase, all nodes do not forward the message replicas until they meet the destination node. (6) SimBet [31]: The messages are forwarded based on the SimBet utility. (7) ESR [42]: An effective algorithm based on the relationship between nodes. (8) FCNS [43]: An effective algorithm based on the fuzzy routing-forwarding.

### 4.2. System Framework

The system framework of optimized routing in the community-based OppNets consists of three parts: data acquisition, methods, and results, as shown in Figure 3.

In the part of data acquisition, the raw data can be obtained from three ways: open experimental data (e.g., the Infocom 2006 trace datasets), simulation and experimental data (e.g., ONE simulator), and actual data acquisition. These data may be incomplete, noisy, and inconsistent, and the use of data cleaning techniques to correct detected errors and omissions is needed. These improved data were saved to the cache of nodes to use again in the future. Then, three important data (including social features, historical encounters, and distances between nodes) can be gained after feature selection. It should be noted that social features of nodes are static internal and long-term characteristics and only need to be collected once, while historical encounters and distances between nodes are dynamic and need to be updated continually.

In the methods, its main task is to the stated method in Section 3—to construct communities and deploy adaptive nodes to the network. In the result showing, its main task is to the system platform (e.g., the simulator ONE) to adjust the parameters of algorithms and evaluate the performance of the optimized routing.

### 4.3. Results

#### 4.3.1. Influences of Ferry Nodes

This simulation evaluates the influence of the ferry node on the message forwarding by comparing ECRF with ECR and SFWP. From Figure 4 and Figure 5, it was clear that ECRF with ferry nodes had better performance than ECR, and ECRF was also superior to SFWP with the same number of ferry nodes. In the simulation of Infocom 2006, with the growth of the number of messages in the network, the delivery ratio and the average delay in ECRF were stable and maintained at about 65% and 1600 s, respectively, while the delivery ratio in SFWP was reduced by about 52% and average delay decreased about 25%. The delivery ratio in ECR was reduced by about 50% and average delay decreased about 25%. The situation in the synthetic trace was roughly similar. These simulation data are explained by: The ferry nodes aid the improvement of the message forwarding performance, and the degree of influence is related to the location of the ferry node.

#### 4.3.2. Comparison of Community Detection Algorithms

In this simulation, we compared the performance of different algorithms from the aspects of community structure detection and message forwarding.

We used the Infocom 2006 datasets to implement the community structure detection. Figure 6 showed the results that the ECR algorithm was slightly better than the k-clique. In ECR, the community structure was clearer and more accurate, especially in areas of the margin or where the community structure was not obvious. In addition, in the practical application of OppNets, the global attribute of the network (including the topology structure) is difficult to obtain because of the movement of nodes and the lack of continuous end-to-end links. Hence, ECR is more suitable for OppNets.

In the performance of community detection algorithms, Figure 7 showed the results that ECR had the better forwarding performance, especially compared to CR. In the simulation based on Infocom 2006 datasets, the delivery ratio in ECR increased by 23%, the forwarding delay reduced by 26%, and the number of forwarding decreased by 5%. In the simulation based on synthetic contact trace, the delivery ratio in ECR rose by 4%, the forwarding delay decreased by 5%, and the number of forwarding decreased by 5%. From the simulation results, we can draw the conclusion that our proposed community detection algorithm is effective due to the fact that it has taken advantage of the accurate node relationship and the better method of community construction.

#### 4.3.3. Comparison of Routing Algorithms

This simulation is based on several contact tracks to evaluate our proposed ECRF by comparing with several classical routing algorithms. From Table 1 and Table 2, it can be seen that ECRF had the best performance among the routing schemes in different situations. In simulation of Infocom 2006, compared to S-W, SimBet, ESR, and FCNS, ECRF is, respectively, 47%, 30%, 1.4%, and 2.3% higher in the delivery ratio, 38%, 27%, 7.5%, and 8.1% lower in the delivery delay, and 15.8%, 7.7%, 0.8%, and 1% lower in the number of forwarding. In simulation of synthetic trace, ECRF is, respectively, 5.4%, 3.4%, 1.8%, and 2.3% higher in the delivery ratio, 9.4%, 3.6%, 0.9%, and 1.1% lower in the delivery delay, and 17%, 9.1%, 2.9%, and 4% lower in the number of forwarding. Obliviously, the community structure and message ferry can remarkably improve the routing performance in OppNets for mHealth.

## 5. Discussion

Due to the randomness of node mobility and the dependence on the forwarding of relay nodes, OppNets typically have poor routing performance and unsafe data leaks. Thus, we introduce the relationship between nodes, community, and message ferry to assist forwarding decision and relay selection.

(1) Complexity of the proposed algorithm

As we could see, there are mainly three computing processes in our algorithm: The measurement of the relationship between nodes, the construction of the community, and the deployment of ferry nodes. They use mainly their own information of each node in each period, which do not consume much of the network resources to transmit control and maintenance information. The time complexity of three processes are, respectively, *o*(*n*), *o*(*n*^2^), and *o*(*n*). Specially, the time complexity of community detection depends on the scale parameters α, and is close to the linear time if α is small enough.

(2) Effect of parameters

Firstly, in the calculation of the relationship metric, as in Equation (6), there is an adjustment parameter α to control the proportion of historical encounters and social features in the relationship. Clearly, *α* is an important parameter to the accurate measurement of node relationship and the construction of community. In different application cases, *α* can take different values to obtain better network performance. In this paper, its value is 0.5, that is, the influence of historical encounters and social features is the same on the relationship between nodes. Next, the center value of nodes is a vital factor to form community, because a mistake will often lead to a very bad result. In this paper, we use a simple calculation method of the center value, as in Equation (7), in order not to waste precious and rare network resources. Finally, *α* in Equation (9) is another important parameter, which can control the size of the community, and is related to the time complexity of community formation. The smaller the value of α is, the finer the community size is, and the higher the time complexity is.

(3) Limitation of proposed algorithm

To improve the accuracy of the node relationship measurement, we used all attributes of the node, including historical encounters and social features. The first ones, which depend on the physical proximity of nodes, are dynamic and need to be updated every once in a while, whereas the second ones are stable and can be obtained before the deployment of OppNets. On the other hand, we only consider the direct relationship between nodes. In fact, the indirect relationship between nodes also affects the motion of nodes, but it increases the complexity of the system.

(4) Future work

It is reasonable that two nodes without a direct relationship still can have a close indirect relationship, and can contact frequently through a very close node in common. Hence, the indirect relationship between nodes can be introduced to improve the measurement of relationship and the center value of nodes. On the other hand, the communities may overlap. That is, one node is part of some different communities. For example, a man in real life has a group of classmates, relatives, and colleagues, etc. Therefore, this multi-level hybrid community performs better in forwarding messages and is worthy of further study.

## 6. Conclusions

In this paper, we have discussed the application of mHealth based on an adaptive routing Optimization Algorithm in OppNets. To achieve the secure message forwarding, we analyze the relationship between nodes and present the average message forwarding delay as a new metric to select the closely related node as the relay. At the same time, we resort to the improved community structure and ferry nodes, which can be adaptively deployed in the network based on the virtual potential field theory, in order to efficiently transfer messages. In future implementation and application, we will use more practical activities based on different realistic and synthetic traces of node movement to test our point of view. We also plan to explore the relationship between nodes based on big data analysis to study more sophisticated routing schemes.

## Figures and Tables

**Figure 1 sensors-19-01876-f001:**
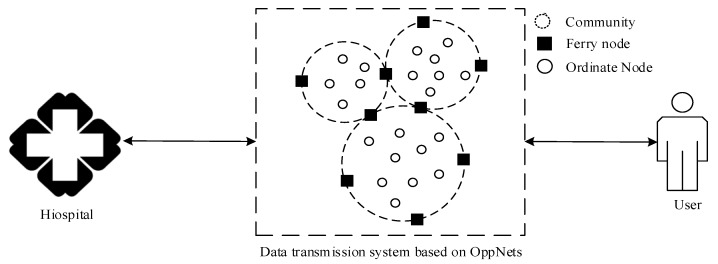
Schematic diagram of the network model that is used in our study.

**Figure 2 sensors-19-01876-f002:**
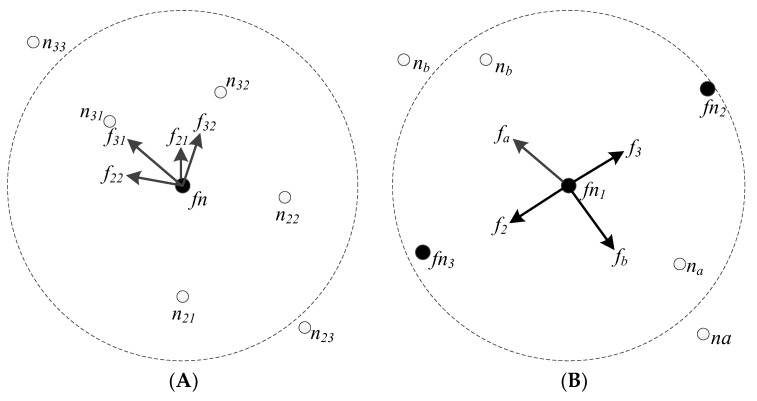
Force diagram of the ferry node presented: (**A**) in a community; and (**B**) between communities.

**Figure 3 sensors-19-01876-f003:**
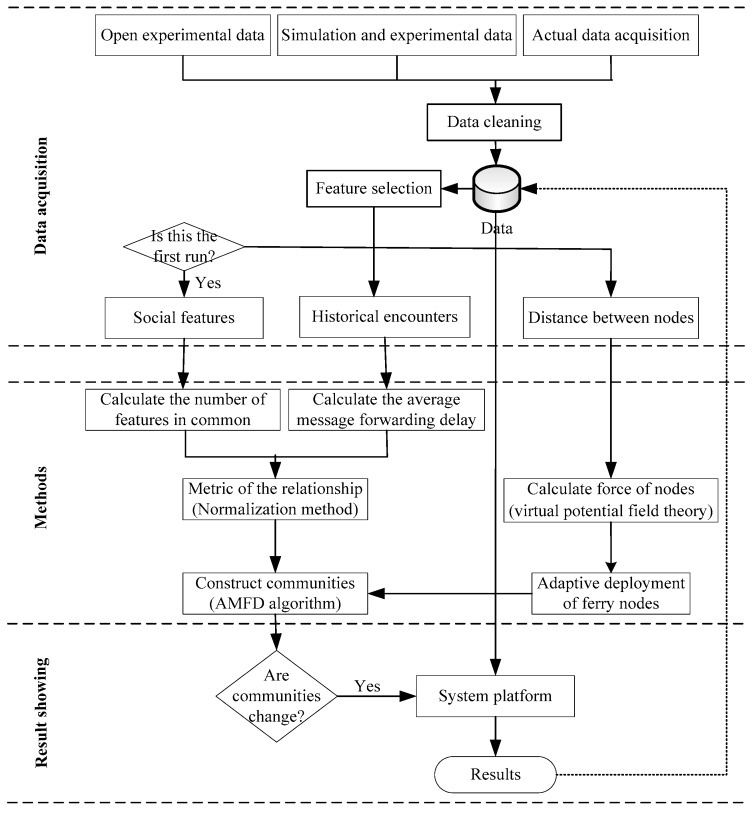
System framework for optimized routing in the community-based Opportunistic Networks.

**Figure 4 sensors-19-01876-f004:**
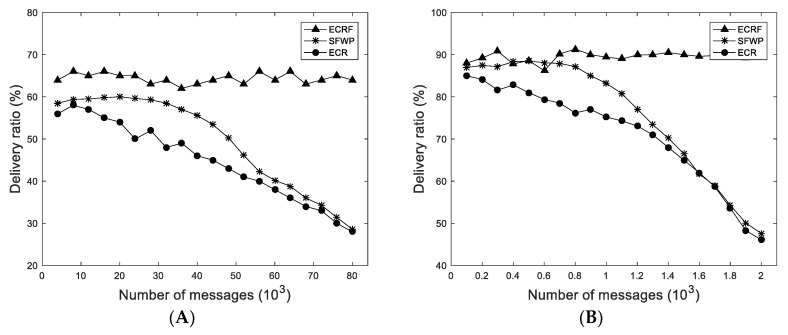
Influences of ferry nodes on delivery ratio: (**A**) In Infocom 2006; and (**B**) in synthetic trace.

**Figure 5 sensors-19-01876-f005:**
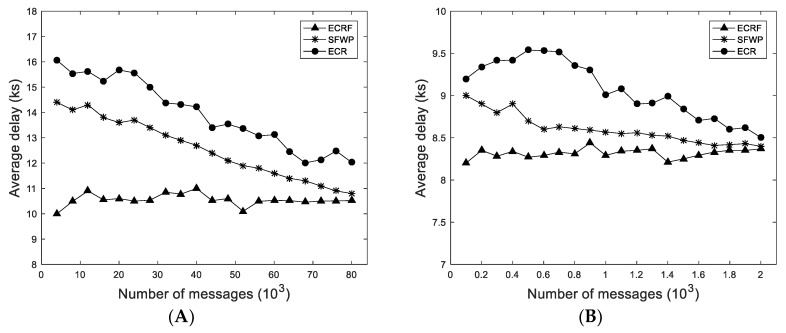
Influences of ferry nodes on average delay: (**A**) In Infocom 2006; and (**B**) in synthetic trace.

**Figure 6 sensors-19-01876-f006:**
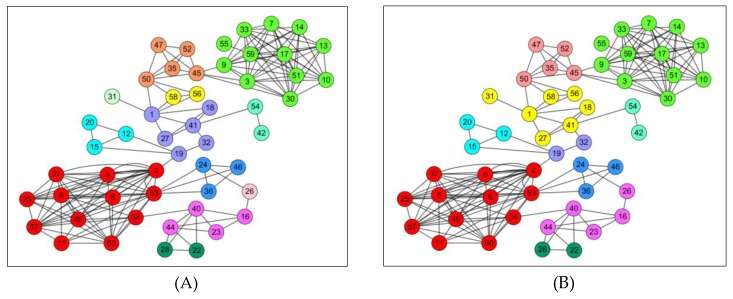
Comparison of community structure detection: (**A**): K-clique: an efficient algorithm for detecting communities in a network; and (**B**) ECR: our proposed community detection algorithm.

**Figure 7 sensors-19-01876-f007:**
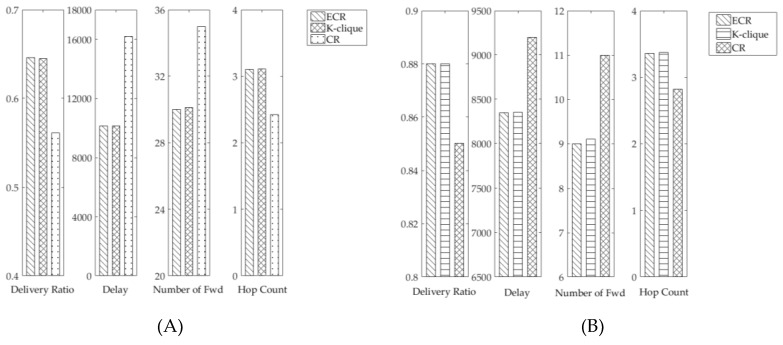
Performance of community detection algorithms: (**A**) In Infocom 2006; and (**B**) in synthetic trace.

**Table 1 sensors-19-01876-t001:** Performance of different routing in Infocom 2006.

Metrics	ECRF ^a^	S-W ^b^	SimBet ^c^	ESR ^d^	FCNS ^e^
Delivery Ratio	0.637	0.432	0.488	0.628	0.623
Delay(s)	9124	14,639	12,346	9860	9924
No. of Fwd	28.84	34.24	31.23	29.06	29.13
Hop count	3.2	2.9	2.5	3.1	3.0

^[a]^ Enhanced community-based routing with ferry node; ^[b]^ Spray-and-wait algorithm; ^[c]^ SimBet algorithm: The messages are forwarded based on the SimBet utility; ^[d]^ Effective algorithm based on the relationship between nodes; ^[e]^ Effective algorithm based on the fuzzy routing-forwarding.

**Table 2 sensors-19-01876-t002:** Performance of different routing in synthetic trace.

Metrics	ECRF	S-W	SimBet	ESR	FCNS
Delivery Ratio	0.877	0.832	0.848	0.861	0.857
Delay(s)	8324	8863	8634	8398	8412
No. of Fwd	9.84	11.84	10.82	10.13	10.25
Hop count	3.2	2.8	2.7	3.0	3.1

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
