# Peer review of "Adaptive Routing Optimization Algorithm in Community-Oriented Opportunistic Networks for Mobile Health"

_sensors, 2019, doi:10.3390/s19081876_

Round 1
Reviewer 1 Report
There are many grammar and spelling errors that need to be fixed before publishing the paper.
The opportunistic routing solutions given as examples in the related work section are pretty old and somewhat irrelevant nowadays. Newer and better solutions exist (based on context data such as social networks, interests, contacts, etc.), that the authors should at least mention.
The acronym for Pocket-Switched Networks is PSN, not SPN.
The BUBBLE Rap paper is written by Pan Hui, where Hui is the last name (so it should be "Hui et al." in the text).
The proposed community detection solution should at least be compared to k-clique by Palla et al., which is the most used solution for opportunistic networks.
How do the authors envision the ferry nodes? What kind of devices are they? They will be under a lot of strain battery-wise (ferrying data from one community to another, probably in a continuous fashion), so they need to have large battery capacities or actually be plugged in (in which case they won't be able to move, though).
How can the ferry nodes move according to the proposed scheme? Are they drones?
The routing algorithm does not really bring anything novel, similar solutions have been proposed over the years.
The solutions that are used for comparison in Section 4 are relatively old and proven to not be exactly optimal.
The mapping of the proposed solution to the area of mHealth is never mentioned. The solution is only theoretical, without real-life applicability.
Author Response
Dear edit in chief:
I am very grateful to your comments for the manuscript entitled “Adaptive Routing Optimization Algorithm in Community-oriented Opportunistic Networks for Mobile Health” (ID: sensors-472898). According with your advice, we amended the relevant part in manuscript. Some of your questions were answered below.
1. The opportunistic routing solutions given as examples in the related work section are pretty old and somewhat irrelevant nowadays. Newer and better solutions exist (based on context data such as social networks, interests, contacts, etc.), that the authors should at least mention.
Response: Thank you for pointing this out. Advanced and better solutions has been added into the revised manuscript, as shown the related work.
Action: On the other hand, to improve routing performance of OppNets, many routing algorithms are proposed from different perspectives, such as Spray and Wait [14], PRoPHET [15]. With the popularization of mobile intelligent terminal, a number of new applications of OppNets have been spawned such as Pocket Switched Networks (PSN) and Vehicular Networks. In these scenarios, the community-based routing algorithms are proposed. Hui et al. [16] described the community structure of PSN and proposed an algorithm named BUBBLE which selects nodes with high centrality value and community members of destination as relay nodes. Jianwei et al. [17]organize nodes into communities on the basis of the contact frequencies and choose more active nodes as the relays in communication. Thomas et al. [18]use group structures for efficient routing in term of the relationship among human society. In [7], Fang et al. used a simple means construct a new cross-group network structure. Jia et al. [19] present a data transmission scheme based on the weight distribution between nodes and communities reconstitution in social opportunistic networks. Zhuanlian et al. [20] propose a community detection algorithm based on low-rank subspace learning, which has a better performance on discriminating the community boundaries. Xingyuan et al. [21] propose a overlapping communities detection algorithm based on vital nodes and initial seeds. Xianhuan et al. [21] propose a non-overlapping community detection algorithm based on the trust relationship between users. In addition, there are some routing algorithms based on the relationship [22], interest [23], context aware [24], etc. Further more, the routing based on message ferry is introduced in some applications of OppNets where nodes are distributed sparsely. Wenrui et al. [25] propose a message ferrying scheme and discuss the use of the ferry routing under the conditions of the constrained number of ferry node. The Optimized Way-points [26] was proposed to determine the mobile path of the ferry node by calculating the encounter probability. Xue et al. [27] select node with highest power as ferry and propose single-ferry and multiple-ferry mechanism. Shangxing et al. [28] formulate the problem of robots ferrying messages to communicate between statically-placed source and sink pairs. Harounabadi et al. [29] use a stigmergic communication for the coordination of ferries and the decision functions based on the local observations of a ferry, and presents an Adaptive On-the-fly Decision maker for Multi Ferry delay tolerant networks.
2. There are many grammar and spelling errors that need to be fixed before publishing the paper.The acronym for Pocket-Switched Networks is PSN, not SPN. The BUBBLE Rap paper is written by Pan Hui, where Hui is the last name (so it should be "Hui et al." in the text).
Response: We were really sorry for our careless mistakes. Thank you for your reminding. The spelling and syntax errors have been checked and corrected. All authors names have been checked carefully.
Action: as shown the related work and References.
3. The proposed community detection solution should at least be compared to k-clique by Palla et al., which is the most used solution for opportunistic networks.
Response: As the Reviewer's good advice, we have added the following comparison of Figure 7. The k-clique scheme is a solution of community detection in social networks and complex networks because it has lower time complexity and higher accuracy, and handle the overlap of community. However, in the opportunistic networks, the global attribute of a network (including the topology structure) is difficult to obtain because of the movement of nodes and the lack of continuous end-to-end link. With this in mind, we present a local community detection algorithm.
Action: as shown the Fig. 7 and 8.
4. How do the authors envision the ferry nodes? What kind of devices are they? They will be under a lot of strain battery-wise (ferrying data from one community to another, probably in a continuous fashion), so they need to have large battery capacities or actually be plugged in (in which case they won't be able to move, though). How can the ferry nodes move according to the proposed scheme? Are they drones?
Response: Thank you for making a very good point here. In our study, the ferry node has the following characteristics: (1) It only undertakes the task of message transmission in the area with sparse nodes (between communities), and the messages in the community is still forwarded by ordinary nodes, so its communication range is not required to be large, which is more than 2 times of ordinary nodes. (2) It can only move when the network changes significantly. As a result, it consumes less energy than the scheme of relying entirely on the ferry node to forward messages, or moving along a fixed path all the time. In our study, the ferry nodes may be a kind of mobile robots, vehicles or drones.
5. The routing algorithm does not really bring anything novel, similar solutions have been proposed over the years.
Response: Thank you for your guidance. The manuscript main focuses on the fellow content: (1) To more accurately measure the relationship between nodes, we define a new metric using the average message forwarding delay with the historical encounters. (2) To avoid the difficulty that the global information of OppNets is too hard to obtain, we present a local community detection algorithm based on the metric. (3) we propose an adaptive deployment of ferry nodes to improve the routing performance in OppNets.
6. The solutions that are used for comparison in Section 4 are relatively old and proven to not be exactly optimal.
Response: Special thanks to you for your good suggestion. We've redone the experiment and rewritten this part. Our consideration is that these solutions (including Spray and Wait, SimBet) are relatively mature in various situations. The former is the better algorithm based on replication and the latter is the better algorithm based on prediction, and they are is the most used algorithms for routing performance comparisons. We are very sorry for our incorrect consideration.
Action: as shown the Table 2 and 3.
Table 2 Performance of different routing in Infocom 2006
Metrics | ECRF | S-W | SimBet | ESR | FCNS |
Delivery Ratio | 0.637 | 0.432 | 0.488 | 0.628 | 0.623 |
Delay(s) | 9124 | 14639 | 12346 | 9860 | 9924 |
No. of Fwd | 28.84 | 34.24 | 31.23 | 29.06 | 29.13 |
Hop count | 3.2 | 2.9 | 2.5 | 3.1 | 3.0 |
Table 3 Performance of different routing in synthetic trace
Metrics | ECRF | S-W | SimBet | SER | FCNS |
Delivery Ratio | 0.877 | 0.832 | 0.848 | 0.861 | 0.857 |
Delay(s) | 8324 | 8863 | 8634 | 8398 | 8412 |
No. of Fwd | 9.84 | 11.84 | 10.82 | 10.13 | 10.25 |
Hop count | 3.2 | 2.8 | 2.7 | 3.0 | 3.1 |
7. The mapping of the proposed solution to the area of mHealth is never mentioned. The solution is only theoretical, without real-life applicability.
Response: Thanks for pointing out. In this study, we introduce OppNets into medical health systems to upload patient physiological data and share medical information, and in order to lighten the load on wireless access networks and reduce the cost pressure on users. As you said, at present the solution is only theoretical, without real-life applicability. We are designing and implementing this solution (including equipment and software) in our laboratory (the ministry of education-china mobile joint laboratory of mobile medical).
We tried our best to improve the manuscript and made some changes in the manuscript. These changes will not influence the content and framework of the paper. We appreciate for Editors/Reviewers’ warm work earnestly, and hope that the correction will meet with approval. Once again, thank you very much for your comments and suggestions。
Reviewer 2 Report
The authors propose a new metric using the average message forwarding delay with the historical encounters rate. The results (gains achieved) are quite interesting nevertheless there are some remarks:
- in opportunistic networks, usually is quite difficult to guarantee a return path or to have the message acknowledgement returning to the source node. How the authors can calculate the delivery ratio? Or is it only possible in a simulation scenario?
- the related works must be updated, the last work is dated on 2013. There is no more recent work since?
- why they do not discuss/compare with their previous work in IEEE Access (Y. Yan, Z. Chen, J. Wu, L. Wang, K. Liu and Y. Wu, "Effective Data Transmission Strategy Based on Node Socialization in Opportunistic Social Networks," in IEEE Access, vol. 7, pp. 22144-22160, 2019.) in the same domain?
- some typos must be corrected.
Author Response
Dear edit in chief:
I am very grateful to your comments for the manuscript entitled “Adaptive Routing Optimization Algorithm in Community-oriented Opportunistic Networks for Mobile Health” (ID: sensors-472898). According with your advice, we amended the relevant part in manuscript. Some of your questions were answered below.
1. in opportunistic networks, usually is quite difficult to guarantee a return path or to have the message acknowledgement returning to the source node. How the authors can calculate the delivery ratio? Or is it only possible in a simulation scenario?
Response: Thank you for pointing this out. It is really true as your suggested that it is quite difficult to guarantee a return path or to have the message acknowledgement returning to the source node. Therefore, Our work, such as the relationship between nodes, the construction of communities and the adaptive deployment of ferry nodes, is only depended on the historical encounters and social features of nodes, does not required calculations of the delivery ratio. To evaluate the performance of the proposed algorithm, it only use in the simulation.
2. the related works must be updated, the last work is dated on 2013. There is no more recent work since?
Response: As the Reviewer's good advice, advanced and better solutions has been added into the revised manuscript, as shown the related work.
Action: On the other hand, to improve routing performance of OppNets, many routing algorithms are proposed from different perspectives, such as Spray and Wait [14], PRoPHET [15]. With the popularization of mobile intelligent terminal, a number of new applications of OppNets have been spawned such as Pocket Switched Networks (PSN) and Vehicular Networks. In these scenarios, the community-based routing algorithms are proposed. Hui et al. [16] described the community structure of PSN and proposed an algorithm named BUBBLE which selects nodes with high centrality value and community members of destination as relay nodes. Jianwei et al. [17]organize nodes into communities on the basis of the contact frequencies and choose more active nodes as the relays in communication. Thomas et al. [18]use group structures for efficient routing in term of the relationship among human society. In [7], Fang et al. used a simple means construct a new cross-group network structure. Jia et al. [19] present a data transmission scheme based on the weight distribution between nodes and communities reconstitution in social opportunistic networks. Zhuanlian et al. [20] propose a community detection algorithm based on low-rank subspace learning, which has a better performance on discriminating the community boundaries. Xingyuan et al. [21] propose a overlapping communities detection algorithm based on vital nodes and initial seeds. Xianhuan et al. [21] propose a non-overlapping community detection algorithm based on the trust relationship between users. In addition, there are some routing algorithms based on the relationship [22], interest [23], context aware [24], etc. Further more, the routing based on message ferry is introduced in some applications of OppNets where nodes are distributed sparsely. Wenrui et al. [25] propose a message ferrying scheme and discuss the use of the ferry routing under the conditions of the constrained number of ferry node. The Optimized Way-points [26] was proposed to determine the mobile path of the ferry node by calculating the encounter probability. Xue et al. [27] select node with highest power as ferry and propose single-ferry and multiple-ferry mechanism. Shangxing et al. [28] formulate the problem of robots ferrying messages to communicate between statically-placed source and sink pairs. Harounabadi et al. [29] use a stigmergic communication for the coordination of ferries and the decision functions based on the local observations of a ferry, and presents an Adaptive On-the-fly Decision maker for Multi Ferry delay tolerant networks.
3. why they do not discuss/compare with their previous work in IEEE Access (Y. Yan, Z. Chen, J. Wu, L. Wang, K. Liu and Y. Wu, "Effective Data Transmission Strategy Based on Node Socialization in Opportunistic Social Networks," in IEEE Access, vol. 7, pp. 22144-22160, 2019.) in the same domain?
Response: As the Reviewer's good advice, we have added the following discuss/compare with their previous work in the revised manuscript, as shown the related work and the section 4.
4. some typos must be corrected.
Response: We were really sorry for our careless mistakes. Thank you for your reminding. The typos have been checked and corrected, as shown the related work and References.
We tried our best to improve the manuscript and made some changes in the manuscript. These changes will not influence the content and framework of the paper. We appreciate for Editors/Reviewers’ warm work earnestly, and hope that the correction will meet with approval. Once again, thank you very much for your comments and suggestions。
Round 2
Reviewer 1 Report
The comments have been addressed satisfactorily.
Author Response
Thank you for your comments concerning our manuscript entitled “Adaptive Routing Optimization Algorithm in Community- oriented Opportunistic Networks for Mobile Health” (ID: sensors-472898). Those kind comments are all valuable and very helpful for revising and improving our paper, as well as the important guiding significance to our researches.
Reviewer 2 Report
The authors tried to address all remarks pointed by the reviewer.
They underlined all the corrections in order to facilitate the second revision.
However there are still important improvement to do:
My third remark, to include a discussion/comparison about the previous work, is not well corrected in the v2. The authors only include the reference [23] with any discussion about it. Why the authors did change the names in the reference [23] ? The correct citation is:
Y. Yan, Z. Chen, J. Wu, L. Wang, K. Liu and Y. Wu, "Effective Data Transmission Strategy Based on Node Socialization in Opportunistic Social Networks," in IEEE Access, vol. 7, pp. 22144-22160, 2019.
doi: 10.1109/ACCESS.2019.2898895
Also, the name of the 3rd author was wrong in version 1? What is the real name?
There is a sentence among the figure 1 and its caption.
Why all the equations are underlined in the v2? They are exactly the same than before, aren't they? The same with figure 3.
The section 4 title is not well formated (centralized).
The quality of figure 4 (v2) is worst than in version 1. It must be redone.
The begin of page 12 there are 2 new figures (colored ones). There are no caption or discussion about them. Afterwards, still in page 12 and page 13 there are two captions (Fig7 and Fig8) without figure. I think the authors did not submit the final version.
The gains achieved by the simulations, described in page 13 (v2), are quite different than in v1. For example (v1):
In simulation of Infocom 2006, ECRF is respectively 47% and 30% higher in the delivery ratio, 38% and 27% lower in the delivery delay, and 5% and 5% lower in the number of forwarding.
(v2): In simulation of Infocom 2006, ECRF is respectively 47%, 30%, 1.4%, and 2.3% higher in the delivery ratio, 38%, 27%, 7.5%, and 8.1% lower in the delivery delay, and 15.8%, 7.7%, 0.8%, and 1% lower in the number of forwarding.
Even that there are two new included algorithms, ie, two more values to compare, the old results must remain the same to S-W and SimBet.
Author Response
Thank you for your comments concerning our manuscript entitled “Adaptive Routing Optimization Algorithm in Community- oriented Opportunistic Networks for Mobile Health” (ID: sensors-472898). Those kind comments are all valuable and very helpful for revising and improving our paper, as well as the important guiding significance to our researches. According with your advice, we amended the relevant part in manuscript. Some of your questions were answered below.
1. My third remark, to include a discussion/comparison about the previous work, is not well corrected in the v2. The authors only include the reference [23] with any discussion about it. Why the authors did change the names in the reference [23] ?
Response: We were sorry for our work. According to your advice, we have revised the manuscript. We should note is that the change in the reference [23] is done because we want to unify the format of the reference.
2. Also, the name of the 3rd author was wrong in version 1? What is the real name?
Response: We were really sorry for our careless mistakes. The name of the 3rd author was wrong in version 1, so we have made the change.
3. There is a sentence among the figure 1 and its caption. Why all the equations are underlined in the v2? They are exactly the same than before, aren't they? The same with figure 3. The section 4 title is not well formatted (centralized). The begin of page 12 there are 2 new figures (colored ones). There are no caption or discussion about them. Afterwards, still in page 12 and page 13 there are two captions (Fig7 and Fig8) without figure. I think the authors did not submit the final version.
Response: We are very sorry for our negligence of the layout of tables and figures. Because the position of tables and figures is fixed, this mess of typography occurs when the previous content changes. We have revised these problems.
4. The quality of figure 4 (v2) is worst than in version 1. It must be redone.
Response: Thank you for your suggestions. The figure has been redone.
5. The gains achieved by the simulations, described in page 13 (v2), are quite different than in v1. For example (v1): In simulation of Infocom 2006, ECRF is respectively 47% and 30% higher in the delivery ratio, 38% and 27% lower in the delivery delay, and 5% and 5% lower in the number of forwarding. (v2): In simulation of Infocom 2006, ECRF is respectively 47%, 30%, 1.4%, and 2.3% higher in the delivery ratio, 38%, 27%, 7.5%, and 8.1% lower in the delivery delay, and 15.8%, 7.7%, 0.8%, and 1% lower in the number of forwarding. Even that there are two new included algorithms, ie, two more values to compare, the old results must remain the same to S-W and SimBet.
Response: Thanks for pointing out. The reason for this is that we did the calculation wrong in the first version. It's worth mentioning that the experimental data are unchanged.